# Off-Gassing of Semi-Volatile Organic Compounds from Fire-Fighters’ Uniforms in Private Vehicles—A Pilot Study

**DOI:** 10.3390/ijerph18063030

**Published:** 2021-03-16

**Authors:** Andrew P. W. Banks, Xianyu Wang, Chang He, Michael Gallen, Kevin V. Thomas, Jochen F. Mueller

**Affiliations:** QAEHS, Queensland Alliance for Environmental Health Sciences, The University of Queensland, 20 Cornwall Street, Woolloongabba, QLD 4102, Australia; x.wang18@uq.edu.au (X.W.); c.he@uq.edu.au (C.H.); m.gallen@uq.edu.au (M.G.); kevin.thomas@uq.edu.au (K.V.T.); j.mueller@uq.edu.au (J.F.M.)

**Keywords:** polycyclic aromatic hydrocarbons (PAHs), organophosphorus flame retardants (OPFRs), polybrominated diphenyl ethers (PBDEs), firefighters, uniform, off-gassing

## Abstract

Firefighters’ uniforms become contaminated with a wide range of chemicals, including polycyclic aromatic hydrocarbons (PAHs), organophosphate flame retardants (OPFRs), and polybrominated diphenyl ethers (PBDEs). Laundering practices do not completely remove PAHs, OPFRs, and PBDEs from firefighting uniforms. This residual contamination of firefighting ensembles may be an ongoing source of exposure to firefighters. Firefighters are known to occasionally store firefighting ensembles in private vehicles. This study aimed to assess whether a firefighting uniform in a vehicle could act as a source for PAHs, OPFRs, and PBDEs to vehicle users. The shell layers of four laundered firefighting uniforms were sampled non-destructively. Three of these uniforms were heated in a laboratory oven (40, 60, and 80 °C) while the fourth was placed in a private vehicle on a summer day and off-gassing samples were collected from the uniforms. The off-gassing results for PAHs and OPFRs were relatively consistent between laboratory oven and the in-vehicle sample with ∑_13_ PAHs in off-gas ranging from 7800–23,000 ng uniform^−1^ day^−1^, while the ∑_6_ OPFRs off-gassed was an order of magnitude lower at 620–1600 ng uniform^−1^ day^−1^. The off-gassing results for PBDEs were much lower and less consistent between the experiments, which may reflect differences in uniform history. Currently, there is limited understanding of how PAHs, OPFRs, and PBDEs off-gassed from firefighting uniforms influence firefighter exposure to these chemicals. These findings suggest that firefighting ensembles off-gassing in private vehicles could be a relevant source of PAHs, OPFRs, and PBDEs that contributes to firefighters’ exposure and that this warrants further investigation.

## 1. Introduction

Firefighters are exposed to a range of chemicals while attending fire scenes, including polycyclic aromatic hydrocarbons (PAHs), organophosphate flame retardants (OPFRs), and polybrominated diphenyl ethers (PBDEs). The exposure to PAHs is the result of partial combustion of organic materials. OPFRs and PBDEs are used as flame retardants, in a wide range of products and materials, and can be released from these materials during combustion [1,2]. After firefighters have attended fire scenes, the clothing and equipment worn to protect themselves inevitably become contaminated with residues from the fire [3,4,5].

Laundering practices are ineffective in completely removing PAHs, OPFRs, and PBDEs from firefighting uniforms [6,7,8,9]. It has been suggested that firefighting ensembles could potentially be a vector for exposure to firefighters, with storage conditions potentially determining the exposure pathway to firefighters [10,11]. It has also been shown that firefighting uniforms may subsequently contribute to elevated semi-volatile organic compounds (SVOC) concentrations in dust at the locations where this gear is stored [12].

Australian firefighters are known to occasionally store firefighting ensembles in private vehicles, often in anticipation of being called in while off-duty during summer and fire seasons. This is not purely an Australian phenomenon; with it being reported that career and volunteer firefighters in the United States also store firefighting uniforms in private vehicles [13,14]. In the United States, PAHs in surface wipes from fire engines and firefighters’ vehicles were of similar concentrations with the storage of firefighting uniforms in vehicles hypothesised as the source of these PAHs [15].

There is concern that firefighting uniforms stored in private vehicles may present a source of exposure to off-duty firefighters as well as other passengers that regularly use their vehicles, such as family members. Elevated heat inside a private vehicle may further increase the rate at which SVOCs are transported from firefighting uniforms into private vehicles. The aim of this study was to assess whether a used and laundered firefighting uniform left in a vehicle may act as an emissions source for selected SVOCs. It also aimed to assess how changes in temperature affect the rate of off-gassing.

## 2. Materials and Methods

### 2.1. Experimental Overveiw

The outer shell layers of four structural firefighting uniforms were sampled for PAHs, OPFRs, and PBDEs. Three of these uniforms were heated in a laboratory oven at 40, 60, and 80 °C, while the fourth was placed in a private vehicle. The uniforms were sampled for PAHs, OPFRs, and PBDEs that were off-gassed from them for 24 h (see Section 2.3. Sample Collection). The temperature in the vehicle was monitored over this time period using a temperature data logger Digitech QP6014. Sampling was conducted in January 2020.

### 2.2. Uniform Selection

Four sets of structural firefighting uniforms were used in this study. Three uniforms had a history of being used for thermal testing use at a firefighter training facility. These three uniforms were placed in the ovens at 40, 60, and 80 °C. The fourth uniform was collected from an active fire service member who had recently attended a house fire. This uniform was used for in-vehicle sampling. All four uniforms were the same make and model, although the comprehensive history and time in service of each uniforms was not known. The uniforms had been laundered by fire services industrial laundering contractor since their last use.

### 2.3. Sample Collection

#### 2.3.1. Uniforms

The samples were collected from the outer shell layer of four sets of structural firefighting jackets and trousers prior to off-gassing sampling. Banks et al. previously described the technique used to sample these uniforms [9]. The shell layer was placed onto a filtration manifold (50 mm diameter) and 50 mL of isopropanol was drawn though the fabric at a rate of 1 mL second^−1^. The shell layer extracts from the jacket and trousers in each sample set were combined. The sample extracts were then spiked with internal standards (500 ng D_10_-Phe, 200 ng D_10_-Flu, 50 ng each of D_12_-Chr, D_12_-BbF, D_12_-BaP, D_12_-I123cdP and D_12_-BghiP, 10ng each of D_18_-TCIPP, D_15_-TPhP and D_27_-TNBP, and 1ng of ^13^C_12_-PBDE mixture). The sample extracts were taken to near-dryness and made up in 50 µL of recovery standard (10 ng ^13^C_12_-BDE 77) in isooctane. The results were converted to ng g^−1^ while using the measured density of the shell layer of 229 g m^−2^.

#### 2.3.2. Off-Gassing

Sets of structural firefighting jackets and trousers were placed into a polypropylene bag. The bag contained an 8mm sampling port on one side of the bag and a separate 8 mm inflow port on the other side of the bag. The inflow port was connected to an XAD-2 (~20 g) pre-treatment to remove PAHs, OPFRs, and PBDEs from the ambient air before being drawn into the polypropylene bag. The sampling port was connected to a stainless-steel sampling probe, which was attached to a low-volume active air sampler (LSAM-100 (SAICI Technology Co., Ltd., Dalian, China) operating at 0.22 m^3^ h^−1^ that drew gas through XAD-2 (5g), where PAHs, OPFRS, and PBDEs are trapped. The collected sample included the XAD-2, and the solvent rinses from tubing and all of the glassware in the sampling train. After sample collection, the sampling train was stored at −20 °C until analysis.

Prior to extraction, the samples were spiked with internal standards (500 ng D_10_-Phe, 200 ng D_10_-Flu, 50 ng each of D_12_-Chr, D_12_-BbF, D_12_-BaP, D_12_-I123cdP and D_12_-BghiP, 10 ng each of D_18_-TCIPP, D_15_-TPhP and D_27_-TNBP, and 1ng of ^13^C_12_-PBDE mixture). The sample was extracted using 50 mL of 1:1 acetone:n-hexane solution in an ultra-sonic bath for 60 min. The solvent was removed and this was repeated with a further 50 mL of 1:1 acetone:n-hexane. These solvent extracts were taken to near-dryness and made up in 1 mL of DCM.

The sample extracts were then cleaned up by gel permeation chromatography (GPC), while using a Shimadzu LC-20AC (Shimadzu, Kyoto, Japan) system coupled with an EnvirogelTM GPC Guard Column 4.6 × 30 mm (Waters), an EnvirogelTM GPC Cleanup Column 19 × 300 mm (Waters), and a Shimadzu FRC-10A fraction collector. The mobile phase solvent was DCM, pumped at a flow rate of 5 mL min^−1^. 500 µL of the filtered DCM extract was injected onto the column. The sample was collected from 8.33 until 16.32 min. The collected fraction was then blown down to near-dryness and reconstituted in 50 µL of recovery standard (10 ng ^13^C_12_-BDE 77) in isooctane.

### 2.4. Analysis

The analysis methods used have been previously described in-depth [10,16,17]. In summary, the extracts were analysed using a TRACE GC Ultra (Thermo Fisher Scientific, Waltham, MA, USA) coupled to a TSQ Quantum XLS (Thermo Fisher Scientific, Waltham, MA, USA) triple quadrupole mass spectrometer that was equipped with a TriPlus Autosampler. A DB-5MS column (30 m × 0.25 mm i.d.; 0.25 µm film thickness, J&W Scientific) was used for separation. The oven temperature was programmed as follows: Initial temperature was 80 °C for 2 min and increased to 180 °C at 20 °C min^−1^ and held for 0.5 min, then to 300 °C at 10 °C min^−1^ and held at this temperature for 5 min. The total run-time was 25 min at constant flow rate of 1.0 mL min^−1^. The volume injected was 1.0 μL, in splitless mode. The QqQ mass spectrometer was operated inelectron ionization(EI) mode using the multiple reactions monitoring (MRM) mode with an emission current set at 20 μA. Table 1 presents the list of target compounds analysed.

### 2.5. Quality Assurance and Quality Control

Field blank samples were prepared and analysed alongside the uniforms (*n* = 3), and off-gassing samples (*n* = 3). For the uniform samples, glass fibre filters (Whatman) were used as field blanks. For the off-gassing samples, the field blanks consisted of 5g of XAD-2, loaded into the air sampler, before being removed. Subsequently, the sampling train was solvent rinsed, as per sample collection. The blank samples were extracted and analysed alongside the sample batches. Method detection limits (MDLs) were defined as the average blank concentrations plus three times their standard deviations (SDs). Appendix A list the MDLs for the individual chemicals in each experiment.

## 3. Results and Discussion

### 3.1. Temperature Profile Inside Vehicle Cabin

The average temperature inside the vehicle cabin over 24 h of sampling was 39.7 °C, with a minimum temperature of 25.7 °C and a peak of 75.3 °C (Figure 1).

### 3.2. Concentrations of PAHs, OPFRs and PBDEs in the Shell Layer of Firefighting Uniforms Profile Inside Vehicle Cabin

The concentrations of PAHs, OPFRs, and PBDEs measured in firefighting uniforms are summarised in Figure 2. The individual chemicals concentrations are presented in the Appendix A. All PAHs, TPhP, EHDPP, TEHP, and TMPP were above the MDLs for all analysed uniforms. TBOEP was above the MDL in three of the four uniforms sampled. BDE 100 and 153 were detected above the MDL, each in a single uniform that had previously been used in thermal testing. BDE 47, 99, 100, 154, and 183 were all above the MDL in the shell layer of the active firefighter’s uniform. TDCIPP and BDE 28 were below MDLs in all the analysed uniforms.

The concentrations of ∑_13_ PAHs in the shell layer of uniforms ranged from 360–570 ng g^−1^, Phe and Pyr on average, contributing to 32% and 17% of this, respectively. Concentrations of ∑_6_ OPFRs in the shell layer of uniforms were between 1200–2800 ng g^−1^ with the highest concentrations being of TPhP and EHDPP. On average, they contributed 67% and 31% of the measured OPFR concentration, respectively. The only PBDEs in the shell layer of uniforms that had been previously used for thermal testing were BDE-100 and BDE 153 that were detected in different uniforms at 0.92 and 0.72 ng g^−1^, respectively. The concentration of ∑_7_ PBDEs detected in the shell layer of the firefighting uniform used by the active fire service member was 550 ng g^−1^, with BDE-99 and BDE-47 contributing to 51% and 38% of this concentration, respectively.

While the PAH and OPFR profiles were similar in all the uniforms, there was a difference in the PBDEs profile between the uniform collected from a fire service member and uniforms that had been used previously used for thermal testing. Unlike the fire service member’s uniform, the uniforms that were used for thermal testing may not have been exposed to PBDEs in sufficient quantities to be measured; this is likely due to the emission from the fires that these uniforms had attended may not have contained PBDEs.

### 3.3. Rate of PAHs, OPFRs and PBDEs Off-Gassing from Firefighting Uniforms

Figure 2 summarises the rate at which PAHs, OPFRs, and PBDEs off-gassed from firefighting uniforms. The results for individual chemicals are presented in the Appendix A. TPhP, TEHP, TMPP, BDE 47, all PAH compounds, with the exception of Pyr, were detected above the MDLs in all collected off-gassing samples. Pyr and TDCIPP were above the MDLs in three of the four samples. EHDPP was above the MDL for two samples, while TBOEP was only above the MDL in a single sample. BDEs 27, 99, 100, 153, 154 and 183 were below the MDL in all collected off-gassing samples.

The rate of ∑_13_ PAHs off-gassing from the uniforms ranged from 7800–23000 ng uniform^−1^ day^−1^. Phe and Ant, on average, contributed to 77% and 9.9% of this, respectively. The rate at which ∑_6_ OPFRs off-gassed were between 620–2800 ng uniform^−1^ day^−1^, with the highest concentrations being from TPhP, which, on average, contributed to 83% of the OPFR emissions. BDE-47 off-gassed from the uniforms between 0.65–53 ng uniform^−1^ day^−1^.

### 3.4. Exposure Implications

The majority of previous research on SVOCs in firefighting uniforms has suggested that dermal exposure from firefighting uniforms may contribute to firefighters’ overall occupational exposure. This study indicates that the off-gassing of chemical contamination from protective clothing into private vehicles may also be a source of chemicals contributing to firefighters’ exposure. This is important, due to the potential ongoing exposure that firefighters may have to chemicals outside of fire scenes. This exposure may be through off-gassing chemicals being directly inhaled or through dermal exposure once they have deposited on surfaces inside vehicles. The ∑_13_ PAHs, ∑_6_ OPFRs, and ∑_7_ PBDEs off-gassed from a firefighter’s uniform in a vehicle was at rates of 12,000, 1200, and 53 ng uniform^−1^ day^−1^, respectively. It has been previously proposed that there is a build-up of PAHs in private vehicles where firefighting uniforms are stored [15]. The same is likely true for OPFRs and PBDEs. Currently, there is limited understanding of how PAHs, OPFRs, and PBDEs that off-gas from firefighting uniforms contribute to overall firefighter exposure. Firefighting uniforms contaminating private vehicles may also have exposure implications to other regular users of the vehicles, such as family members. This may be a significant and ongoing source of exposure that necessitates further investigation.

### 3.5. Factors That May Impact the Emissions

It was hypothesised that warmer temperatures would induce a higher rate of SVOC off-gassing from the contaminated firefighter uniforms. In our study, the temperature did not appear to affect the rate at which PAHs, OPFRs, and PBDEs off-gassed from uniforms. (Appendix A). Instead, the rate at which chemicals off-gassed from firefighting uniforms appeared to be mainly driven by the vapor pressure of individual chemicals. Figure 3 and Appendix A present a moderate relationship between the vapor pressure and ratio of concentration in the shell layer to the rate of off-gassing.

### 3.6. Limitations and Perspectives

The sample size of four uniforms used in this study was relatively small and, therefore, a high level of uncertainty may be associated with the study. The sampling method used to measure the off-gassing of PAHs, OPFRs, and PBDEs from firefighting uniforms only sampled chemicals in the gaseous phase and, thus, neglected compounds bound to particles. This may underestimate the amount of target compounds emitted into vehicles. The unknown efficiency of the non-destructive sampling in the removal of PAHs, OPFRs, and PBDEs from shell layers is a further limitation of this study. However, all shell layer samples collected would be equally affected by this. Finally, we were unable to collect samples from thermal liners or moisture barriers inside firefighting jackets and trousers due to this non-destructive method. The emission of SVOCs from these layers, if present, are likely to influence the rate of off-gassing. This could be one of the factors contributing to why a relationship between the temperature and rate of off-gassing was not observed.

## 4. Conclusions

This study reinforces the need for ongoing research into the removal of SVOCs from firefighters’ uniforms. The findings from this study suggest that firefighting ensembles off-gassing in private vehicles could be a significant source of chemicals that contribute to firefighters’ exposure to PAHs, OPFRs, and PBDEs. Ideally, modified washing techniques that adequately remove SVOCs from firefighting would remove uniforms as a source of this exposure. Until then, there is a need to have a greater understanding of the effectiveness of different storage methods at containing SVOCs off-gassing from firefighters’ uniforms in vehicles.

## Figures and Tables

**Figure 1 ijerph-18-03030-f001:**
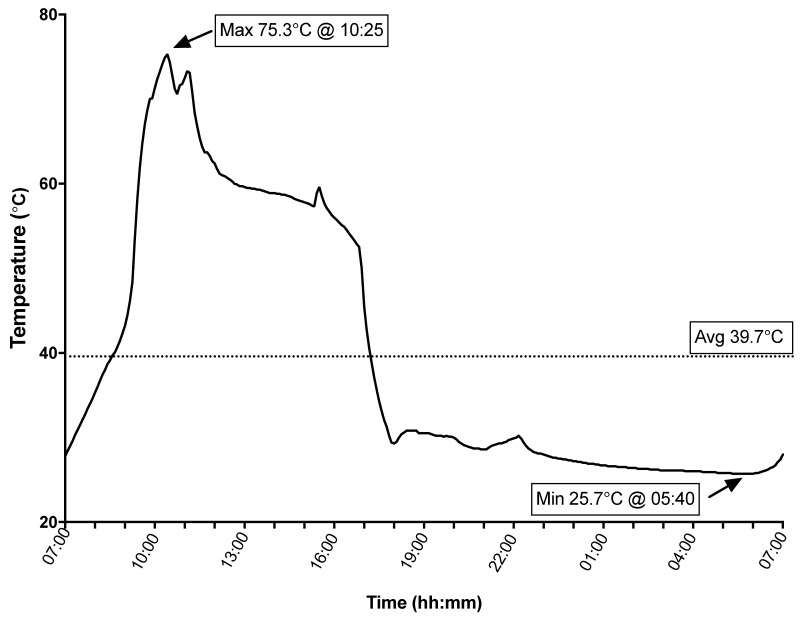
Temperature profile inside vehicle cabin.

**Figure 2 ijerph-18-03030-f002:**
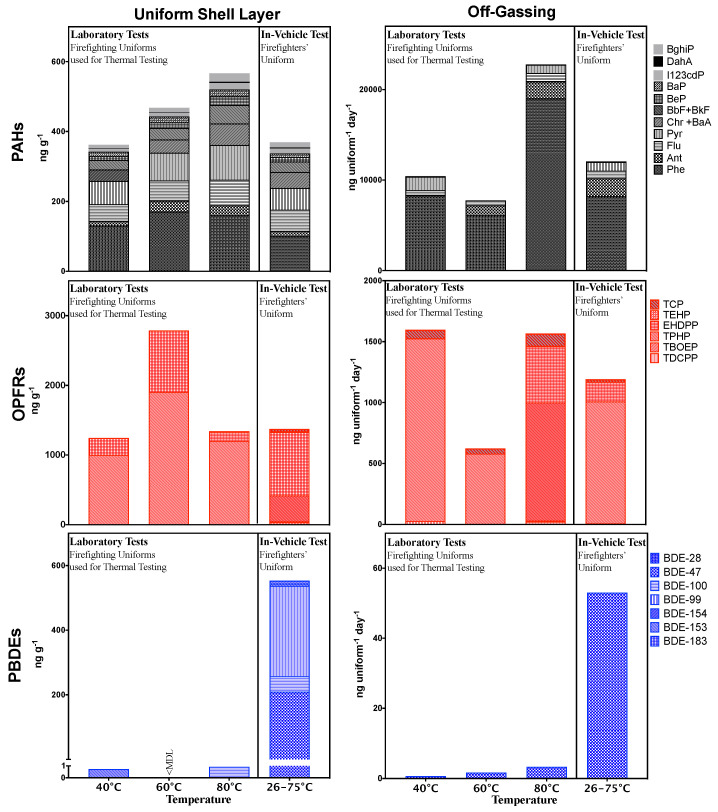
Concentrations of polycyclic aromatic hydrocarbons (PAHs), organophosphate flame retardants (OPFRs), and polybrominated diphenyl ethers (PBDEs) in structural firefighting uniforms (ng g^−1^) and rate of off-gassing (ng uniform^−1^ day^−1^). (< MDL = 0).

**Figure 3 ijerph-18-03030-f003:**
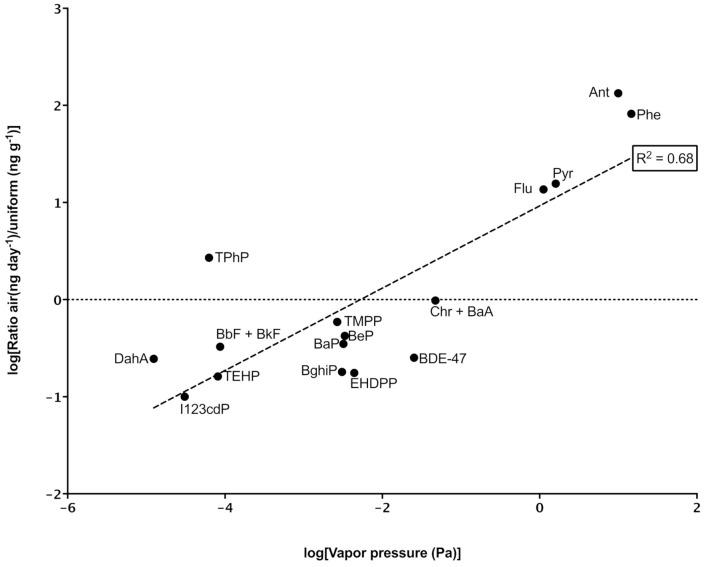
Relationship between vapor pressure and the rate at which semi-volatile organic compounds (SVOCs) off-gassed from a structural firefighting uniform in a vehicle.

**Table 1 ijerph-18-03030-t001:** List of targeted compounds.

Chemicals	Abbreviation	CAS Number
PAHs		
Phenanthrene	Phe	85-01-8
Anthracene	Ant	120-12-7
Fluoranthene	Flu	86-73-7
Pyrene	Pyr	129-00-0
Chrysene	Chr	218-01-9
Benz[a]anthracene	BaA	56-55-3
Benzo[b]fluoranthene	BbF	205-99-2
Benzo[k]fluoranthene	BkF	207-08-9
Benzo[e]pyrene	BeP	192-97-2
Benzo[a]pyrene	BaP	50-32-8
Indeno[1,2,3-c,d]pyrene	I123cdP	193-39-5
Dibenzo[a,h]anthracene	DahA	200-181-8
Benzo[ghi]perylene	BghiP	191-24-2
OPFRs		
Tris(1,3-dichloroisopropyl) phosphate	TDCIPP	13674-87-8
Tris(2-butoxyehyl) phosphate	TBOEP	78-51-3
Triphenyl phosphate	TPhP	115-85-6
2-ethylhexyl diphenyl phosphate	EHDPP	1241-94-7
Tris(2-ethylhexyl) phosphate	TEHP	78-42-2
Tris(2-methylphenyl) phosphate	TMPP	78-30-8
PBDEs		
2,4,4′-tribromodiphenyl ether	BDE-28	41318-75-6
2,2′,4,4′-tetrabromodiphenyl ether	BDE-47	5436-43-1
2,2′,4,4′,5-pentabromodiphenyl ether	BDE-99	60348-60-9
2,2′,4,4′,6-pentabromodiphenyl ether	BDE-100	189084-64-8
2,2′,4,4′,5,5′-hexabromodiphenyl ether	BDE-153	68531-49-2
2,2′,4,4′,5,6′-hexabromodiphenyl ether	BDE-154	207122-15-4
2,2′,3,4,4′,5′,6-heptabromodiphenyl ether	BDE-183	207122-16-5

## Data Availability

The data presented in this study are openly available in the Appendix A document.

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
