# Peer review of "Off-Gassing of Semi-Volatile Organic Compounds from Fire-Fighters’ Uniforms in Private Vehicles—A Pilot Study"

_ijerph, 2021, doi:10.3390/ijerph18063030_

Round 1
Reviewer 1 Report
Review ijerph-1134587
The manuscript refers to the exposure of firefighters to semi-volatile organic compounds arising from storage of firefighting uniforms in vehicles as laundering practices are not completely effective according to removing of harmful substances. The manuscript is interesting and up to date. I would suggest some improvements mentioned below.
Major remarks
Results and discussion. I would suggest to introduce/develop the paragraph related to potential risk of inhalation and dermal contact with harmful substances off-gasing from firefighting uniforms and to point why/how it is important in the context pf public health.
According to lines 230-231 “There is also a need to have a greater understanding of methods to store firefighting uniforms to prevent the contamination of private vehicles” I would also suggest to describe more precisely what should be investigated in the future and with methods as well as why in the context of potential exposure for passengers of firefighters’ vehicles with semi-volatile compounds.
Minor remarks
Line 61. Were firefighting uniforms of the same type, age, usage, etc. prior to comparing results from different temperatures in laboratory tests and in car experiment? Please discuss in the manuscript.
Line 64. “see below”. I would suggest to rephrase to point the specific subsection.
Line 70. Please correct the symbol to °.
Lines 186-187. “This study indicates that the off-gassing of chemical contamination from protective clothing may also be a contributing exposure pathway”. In the risk assessment methodology exposure pathways are ingestion, inhalation, and dermal contact. Please rephrase the sentence as off-gassing itself cannot be the exposure route but rather the source of chemical contamination in specific exposure route (mentioned in previous sentence).
Line 194. those vehicles?
Lines 221-225. I believe this part was not deleted from the template. Please check.
Line 229. Please see the comment for lines 186-187 and correct accordingly.
Reviewer 2 Report
The case report “Off-Gassing of Semi-Volatile Organic Compounds from a Fire-fighters’ Uniforms in Private Vehicles – A Pilot Study” describes the presence of polyaromatic hydrocarbons and common flame retardants in fire fighters clothing as a possible hazard when stored in private cars. The topic of the case report is worth investigating because it describes a less considered hazard for firefighters and also their families. The described analytics and their discussed limitations are sound. The estimated scenarios (e.g. temperature profile in the car) are plausible. However, some aspects need to be improved before publication.
The biggest weakness of the report is the presentation of the results in Figure 2. Even at the highest zoom it is very hard to read. I recommend using a larger font and align the diagrams vertically. The discrimination between the different chemicals is very hard in case of the PAH. The authors should consider to transform the diagram into a table.
In the conclusion section I am missing some further thoughts with regard to exposure prevention. Technically, the described issue can be easily solved by a modified washing procedure or by protection/storing bags when contaminated clothes are stored at elevated temperatures. These simple recommendations should be added to the report.
Minor comments:
The authors forgot to remove section 4 when merging the chapters to “Results and Discussion”. Section 4 contains text from the manuscript template.
I recommend checking the manuscript for typos (e.g. “concertation” in line 151).
Some units have changed due to typesetting: In case of “ng.g-1” I recommend removing the “.” or use a “*” instead.
Round 2
Reviewer 1 Report
All of my remarks were sufficiently answered in the revised version of the manuscript.